# SURINPARK: Safinamide for Urinary Symptoms in Parkinson’s Disease

**DOI:** 10.3390/brainsci11010057

**Published:** 2021-01-06

**Authors:** Ana Gómez-López, Arantxa Sánchez-Sánchez, Elena Natera-Villalba, Victoria Ros-Castelló, Álvaro Beltrán-Corbellini, Samira Fanjul-Arbós, Isabel Pareés Moreno, José Luis López-Sendon Moreno, Juan Carlos Martínez Castrillo, Araceli Alonso-Canovas

**Affiliations:** 1Neurology Department, Hospital Universitario Ramon y Cajal, 28034 Madrid, Spain; arantxa212@gmail.com (A.S.-S.); elena.natera.v@gmail.com (E.N.-V.); victoriaroscastello@gmail.com (V.R.-C.); beltran_corbellini@hotmail.com (Á.B.-C.); sfanjul@hotmail.com (S.F.-A.); isabel.parees@salud.madrid.org (I.P.M.); joselopezsendon@hotmail.com (J.L.L.-S.M.); jcmcastrillo@gmail.com (J.C.M.C.); aracelialonsocanovas@gmail.com (A.A.-C.); 2Movement Disorder Unit, Neurology Department, Hospital Universitario Ramon y Cajal, 28034 Madrid, Spain

**Keywords:** safinamide, urinary symptoms, Parkinson disease, nonmotor fluctuations

## Abstract

Background: Urinary symptoms are common, disabling and generally unresponsive to treatment in Parkinson´s disease (PD). Safinamide is approved as an add-on therapy to levodopa to improve fluctuations. Methods: Retrospective analysis of electronic records of nondemented PD patients seen consecutively in a Movement Disorders Unit (November 2018–February 2019). All were assessed with Scale for Outcomes in Parkinson’s disease for Autonomic Symptoms-Urinary subscale (SCOPA-AUT-U) by the attending neurologist, and a month afterwards by an independent researcher blinded to treatment and clinical records in a routine clinical practice setting. Clinical variables were compared among patients who were prescribed safinamide (SA+) for the treatment of motor fluctuations and those with different treatment regimes (SA−). Results: From 169 patients screened initially, 54 were excluded due to severe incontinence, absence of urinary symptoms or previous safinamide treatment. Thirty-five patients were included in SA+ and 79 in SA−. Both groups were comparable in terms of clinical variables, except in basal urinary symptoms, with more severity in the SA+ group. In the follow-up assessment, total SCOPA-AUT-U, as well as urgency, incontinence, frequency and nocturia subscales improved significantly in the SA+ group, while the SA− group remained unchanged. Conclusions: Safinamide could be helpful in the improvement of urinary symptoms in PD.

## 1. Introduction

Parkinson´s disease (PD) is the second most common neurodegenerative disease after Alzheimer´s disease. Degeneration of the pars compacta of the substance nigra precipitates dysfunction of the dopaminergic circuits, although other neuronal circuits like glutamate, noradrenaline or serotonin are also affected [1,2]. Bradykinesia is the cardinal symptom of PD and it may be accompanied by resting tremor and rigidity, but nonmotor symptoms such as sensory disturbances (hyposmia, pain), neuropsychiatric, sleep disorders and dysautonomia are increasingly recognized as symptoms of PD, and they have a paramount impact on the quality of life of these patients [3,4,5,6,7].

Among dysautonomic problems, urinary symptoms are very common across different PD stages and tend to worsen as disease progresses, although some authors have suggested a late decline in frequency [8,9]. The range is wide, from 24 to 96% [10,11]. In the PRIAMO (Parkinson and non motor symptoms) registry, they affected 57% of the studied population. Irritative symptoms, such as nocturia, increased frequency and urgency to micturate, with or without incontinency, are typically associated with PD. Nocturia is the most commonly reported symptom (57–86%, 34% in PRIAMO), followed by increased frequency (32–71%), urgency (32–68%), and urge incontinence (21–40%). The nature of urinary symptoms is probably multifactorial, influenced by motor, cognitive and other dysautonomic aspects [4,12,13,14,15].

As PD affects mainly the elderly, urologic comorbidity (benign prostatic hypertrophy in males, effort urinary incontinence in females) is very common, which hinders the understanding of the PD-related urinary changes [11]. Frequently, urinary symptoms are not attributed to PD, not spontaneously reported by patients and not systematically screened for by the attending neurologists [16]. In this regard, the Scale for Outcomes in Parkinson´s disease for Autonomic Symptoms (SCOPA-AUT) is a valid assessment tool for nonmotor PD symptoms, and the urinary subscale (SCOPA-AUT-U) is easy and quick to administer, exploring the main categories of these symptoms in an intuitive fashion [3,7,17,18].

Optimized dopaminergic stimulation does not result in a corresponding improvement in urinary symptoms in clinical practice, which points to nondopaminergic mechanisms at its origin and maintenance. Optimal treatment is unclear, and few not replicated uncontrolled studies have evaluated specific management [12,14,19]. The dopamine agonist rotigotine showed an improvement in bladder capacity in urologic assessments of 20 patients three months after the drug initiation [20]. Another study about the D1-D2 agonist apomorphine showed that the acute administration of the drug improved bladder emptying and detrusor hyperactivity [21,22]. A brief study on 20 patients showed an improvement on urodynamic values and urologic questionnaire scores with rasagiline [23]. Studies on levodopa yielded conflicting results, both for and against a positive effect on urinary symptoms [24,25].

Safinamide is a reversible, selective, monoamine oxidase b inhibitor (MAO-B-i) and glutamate modulator with therapeutic indication as an add-on to levodopa in fluctuating PD patients [15]. At the dose of 50 mg, safinamide increases the availability of dopamine by inhibiting MAO-B. At higher doses (100 mg), safinamide also blocks voltage-dependent calcium and sodium channels and inhibits glutamate release at overactive synapses [26,27]. Interestingly, several authors have also suggested a possible effect of safinamide in nonmotor symptoms, such as pain and mood [28,29].

The aim of our study was to assess the possible clinical effect of safinamide, used within the label as an additional therapy to levodopa for motor fluctuations, on improving the urinary symptoms of PD patients in daily clinical practice.

## 2. Materials and Methods

We retrospectively collected data from a cohort of unselected consecutive patients with PD followed up at the Movement Disorder Unit of our tertiary University Hospital, who completed routine clinical outpatient assessments between 1st November 2018 and 28th February 2019 (baseline visit, T0). Exclusion criteria were previous treatment with safinamide, severe cognitive impairment and severe urinary incontinence (use of pads 24 h) or retention (bladder catheterization).

Demographic and clinical variables like age, sex, duration of the disease, motor fluctuations, Hoehn and Yahr stage [30], cognitive status [31], urologic comorbidity and associated urological treatments were systematically registered in clinical records by the attending neurologist. The scale of urinary symptoms Scale for Outcomes in Parkinson’s disease for Autonomic Symptoms-urinary (SCOPA-AUT-U) (Appendix A) was also administered as a part of the regular clinical assessment [3,7,17,18].

Changes in treatment regimes for PD were also recorded at the baseline T0. The addition of safinamide was made in an open-label fashion, within drug labeled use, with the aim to improve motor fluctuations as an add-on therapy to levodopa. If safinamide was prescribed to a patient taking MAO-B-i (rasagiline or selegiline), these were withdrawn when safinamide was started.

At least one month after the routine visit (T0), an independent neurologist, blinded to the treatment status and clinical records, performed a telephone assessment (T1) to evaluate urinary symptoms (SCOPA-AUT-U) and eventual adverse events. For the analysis, patients were distributed in two groups depending on the treatment decisions made in T0: Safinamide addition (SA+) or different treatment options (SA−), such as increasing doses of levodopa, pramipexole, ropinirole, rotigotine, entacapone, opicapone, selegiline, rasagiline, amantadine or apomorphine, or keeping the previous regime unchanged.

Baseline demographic clinical variables and specifically basal SCOPA-AUT-U scores (total and subscales scores), as well as the number of nocturia episodes whenever present, were analyzed and compared between SA+ and SA− groups. SCOPA-AUT-U scores (total and subscales scores) change between baseline (T0) and follow-up (T1) visits were compared within groups. The study obtained the approval of the Hospital’s Institutional Ethics Board Committee (no. 206/19).

Descriptive statistics, Fisher’s exact test for qualitative variables and Mann-Whitney U for quantitative variables were used in between groups´ comparisons. SCOPA-AUT-U scores (total and subscales scores) change between baseline (T0) and follow-up (T1) visits were compared within groups with the Wilcoxon signed-rank test. Level of significance was set at *p* < 0.01. G-Stat 2.0 software was used for statistical analyses.

## 3. Results

### 3.1. Patient Characteristics

An initial sample of 169 patients was included, of which 16 patients were excluded because of difficulty in quantifying urinary symptoms or serious urological problems, and 13 because of the absence of urinary symptoms. A total of 26 patients were excluded because they were already on treatment with safinamide 100 mg/daily at T0. During follow-up, three patients of safinamide group (SA+) stopped treatment because of adverse events (confusion and drowsiness) and one patient without safinamide (SA−) was lost to follow-up. Finally, 110 patients were included in the analysis: 32 with safinamide (SA+) and 78 without safinamide (SA−) (Table 1).

Clinical and demographic variables were similar in both groups (Table 1), although a trend towards longer disease duration and Hoehn and Yahr stage was observed in SA+ group. Urological comorbidities and urological treatment were common in both groups, benign prostatic hyperplasia being the most frequent reason for urological comorbidities (16%). However, in the basal SCOPA-AUT-U scores, the safinamide group (SA+) had a higher score compared to the nonsafinamide group (SA+ 9.1 ± 3.1 SD vs. SA− 7.2 ± 3.6 SD; *p* = 0.0098), meaning more severe urinary symptoms at baseline. Concomitant treatments and changes in drug regime in follow-up are detailed in Table 2.

### 3.2. Outcomes

Table 3 details the variation of total score of the SCOPA-AUT-U scale and subscales, yielding a significant change in SA+ group, both in the total score (T0 9.1 ± 3.1 vs. T1 6.6 ± 3; *p* = 0.00004) and in urgency (T0 1.8 ± 1 vs. T1 1.4 ± 1; *p* < 0.01), incontinence (T0 1.3 ± 1 vs. T1 0.8 ± 0.9; *p* = 0.004), frequency (T0 1.7 ± 1.3 vs. T1 1 ± 1; *p* = 0.002) and nocturia (T0 2.7 ± 0.9 vs. T1 2 ± 1.3; *p* = 0.002), as well as a significant decrease in the number of nocturia episodes (T0 2.3 ± 1.4 vs. T1 1.1 ± 1.1; *p* = 0.0001). In contrast, in the nonsafinamide group (SA−), no statistically significant differences were found between the two visits.

### 3.3. Safety Outcomes

Adverse events in SA+ patients (*n* = 3) were mild-moderate confusion and drowsiness but led to drug discontinuation in all three. In the SA−group, no adverse events leading to discontinuation of any new drug or dose variation were recorded.

## 4. Discussion

Nonmotor symptoms are common in all stages of PD, impact quality of life and there is a high need of efficacious, evidence-based treatments [3,4,5]. Urinary symptoms are especially complex from a physiopathological point of view, and have a functional impact on daily activities, social stigma and quality of sleep, among others [32]. In this complex setting, our study provides real clinical practice evidence to suggest a role of safinamide in alleviating these bothersome symptoms.

We performed a retrospective analysis of the clinical records of unselected consecutive patients of our Movement Disorders Unit in a three months period. Safinamide was initiated within label to treat motor fluctuations in 32 patients. We compared this group with the remaining 79 patients (1:2.5 proportion) who were seen in our Movement Disorder Unit in the same period but were not started on safinamide, according to their clinical symptoms and the attending neurologist criterion. Safinamide (SA+) patients showed a trend towards a more advanced disease in their basal features, as expected, as the drug is labeled as an add-on to levodopa in fluctuating disease. Urinary symptoms were strikingly common in the whole sample (only 13 out of the initial 169 patients were excluded due to lack of relevant urinary symptoms), and especially in SA+ group, with significantly higher scores in baseline SCOPA-AUT-U than SA− group. The relationship between the severity of the disease and urinary symptoms is not clear, although a worsening of the symptoms with the appearance of motor complications in the disease is suggested [33,34].

Our analysis showed a generalized improvement of total SCOPA-AUT-U in SA+ group, with a mean 2.5 points (27%) decrease, driven mainly by irritative symptoms: incontinence, urgency and more strikingly, daily frequency (mean decrease of 41%) and nocturia (mean decrease 26%). The lack of effect on tenesmus and weak stream, the least affected subscales in baseline, is probably related to the structural genitourinary origin of these problems, less prone to be affected by oral drugs.

Nocturia is the most common urinary symptom in PD, with important consequences in the quality of sleep and, thus, daily functioning [8]. In our sample, nocturia episodes more than halved by mean (56% reduction), which, despite the lack of a proper multimodal assessment in our study, probably had a more global functional improvement in these patients through sleep quality improvement. Conversely, in many cases the motor problems (akinesia, biphasic dyskinesia, off period dystonia), or different nonmotor problems (pain, insomnia) disrupt sleep and cause nocturia as a consequence of increased nocturnal arousal. Safinamide has been shown to improve sleep quality, probably through a mixed potential effect on several of these areas, although so far nocturia has not been specifically analyzed [28,35]. Certainly, the effect of safinamide in these nonurinary symptoms may have driven or contributed to the improvement of nocturia in our SA+ patients. However, in the Randomized Evaluation of the 24-h Coverage: Efficacy of Rotigotine (RECOVER) study, the overall improvement on sleep quality with rotigotine did not lead to a decrease in nocturia episodes, which may point to a specific effect of safinamide in this area [36].

In fact, the nature of the possible effect of safinamide on urinary symptoms remains speculative. All patients in our sample were started on 100 mg of safinamide, so we could not analyze the effect of a 50 mg dose. We can, however, imply that the nondopaminergic effect of the drug over 50 mg may play a role, in analogy with other nonmotor symptoms such as pain, mood or sleep disorders, in which this effect has been observed more clearly at higher doses [26,27,30,33]. The observation that the control sample (SA−), in which treatment modifications were mostly dopaminergic, did not experience any change in their urinary problems, may support this too. A nondopaminergic effect of safinamide on attention, mood and anxiety, suggested by different studies, could be relevant in this regard, helping a better control of micturition from higher order cortical centers [29,37,38,39,40].

Finally, we ascertained an overall good tolerability of safinamide. Only three patients experienced mild to moderate nonspecific adverse events, such as drowsiness and dizziness, leading to discontinuation of the drug. Regarding motor and other nonmotor effects of the drug, which may have had an impact on the improvement of urinary symptoms, they were not systematically registered in our study, so we cannot provide quantified data on efficacy. We must acknowledge these and other limitations of our study, mainly the open-label and retrospective nature of the design.

Safinamide could not be compared with other individual antiparkinsonian drugs, due to the relatively low number of subjects with each one. In addition, a placebo effect could contribute at least in part to the improvement of urinary symptoms in the short term. However, the drug was not initiated to treat these symptoms, but rather motor fluctuations, and the placebo effect would have affected equally safinamide and the different drug regimes. Last, proper urologic assessments, not feasible in a routine clinical practice setting, would have provided objective measures of the potential urodynamic changes with safinamide.

## 5. Conclusions

Urinary problems were common in our fluctuating PD patients. Our retrospective study suggests a possible benefit of the addition of safinamide 100 mg to their drug regimes, with overall good tolerability and consistent improvement of the different symptoms, especially the irritative and nocturnal. Although the mechanism of this effect remains unknown, we speculate with a nondopaminergic multimodal effect of the drug. Controlled prospective studies are necessary to clarify these findings.

## Figures and Tables

**Table 1 brainsci-11-00057-t001:** Clinical and demographic variables comparison between safinamide group and no safinamide group.

	Safinamide as Add-on Therapy (SA+)	No Safinamide as Add-on Therapy (SA−)	*p*-Value
Age (years) ^a^	74 ± 10	72 ± 10	NS
Male (%) ^b^	20 (62.5%)	45 (57.6%)	NS
Disease duration (years) ^a^	8.2 ± 5.3	6.4 ± 6.3	NS
Hoehn & Yahr stage ^a^	2.1 ± 0.8	1.9 ± 0.7	NS
Basal SCOPA-AUT-U ^a^	9.1 ± 3.1	7.2 ± 3.6	*p* = 0.0098 ^c^
Comorbid urologic condition ^b^	12 (38%)	28 (36%)	NS
Urologic treatment ^b^	6 (19%)	20 (26%)	NS

^a^ Median (SD); ^b^ Number of patients (relative frequency); ^c^ Mann-Whitney test. Abbreviation: NS: no significant; SCOPA-AUT-U: scale for outcomes in Parkinson’s disease for autonomic symptoms in urinary symptoms.

**Table 2 brainsci-11-00057-t002:** Treatment regime in both groups at baseline and follow-up.

	Safinamide as Add-on Therapy (*n* = 32)	No Safinamide as Add-on Therapy (*n* = 79)
**Baseline Treatment (T0)**		
Levodopa (%) ^a^	32 (100%)	61 (78%)
Dopamine agonist (%) ^a^	16 (50%)	31 (40%)
Pramipexole	8 (25%)	12 (15%)
Ropinirole	5 (16%)	7 (9%)
Rotigotine	3 (9%)	12 (15%)
MAO-I (%) ^a^	11 (34%)	6.4 ± 6.3
Rasagiline	10 (31%)	41 (52%)
Selegiline	1 (1%)	1 (1%)
Amantadine ^a^	3 (9%)	3 (4%)
Opicapone ^a^	4 (13%)	3 (4%)
Intermittent apomorphine ^a^	0 (0%)	1 (1%)
Advanced therapies ^a^	2 (6%)	4 (5.1%)
DBS	2 (6%)	3 (4%)
Apomorphine CI	0	1 (1%)
**Follow-Up Treatment (T1)**		
Safinamide ^a^	32 (100%)	0
No change ^a^		49 (63%)
Levodopa/ICOMT ^a^		13 (17%)
Dopamine agonist ^a^		7 (9%)
MAO-I (R/S) ^a^		6 (8%)
Amantadine ^a^		1 (1%)
Combinations (DA+LD) ^a^		2 (3%)

^a^ Number of patients (relative frequency). Abbreviation: T0: baseline visit; MAO-I: monoamine oxidase inhibitor; T1: follow-up visit; (R/S): rasagiline/selegiline; DBS: deep brain stimulation; CI: continuous infusion; COMTI: catechol-O-methyl transferase inhibitor; DA: dopamine agonist; LD: levodopa.

**Table 3 brainsci-11-00057-t003:** Comparison of urinary symptoms in both groups after possible changes in treatment.

	Safinamide as Add-on Therapy (*n* = 32)			No Safinamide as Add-on Therapy (*n* = 78)		
Treatment	Baseline T0	T1 ^a^	*p* Value	Baseline T0	T1 ^a^	*p* Value
SCOPA-AUT-U ^b^	9.1 ± 3.1	6.6 ± 3	*p* = 0.00004	7.2 ± 3.6	7.2 ± 3.6	NS
Urgency ^b^	1.8 ± 1	1.4 ± 1	*p* < 0.01	1.5 ± 1.1	1.5 ± 1.2	NS
Incontinence ^b^	1.3 ± 1	0.8 ± 0.9	*p* = 0.0044	0.8 ± 0.9	0.8 ± 0.9	NS
Tenesmus ^b^	0.9 ± 1	0.6 ± 0.9	NS	0.8 ± 1	0.8 ± 1.1	NS
Weak urinary stream ^b^	0.7 ± 1.2	0.4 ± 0.9	NS	0.4 ± 0.9	0.4 ± 0.9	NS
Frequency ^b^	1.7 ± 1.3	1 ± 1	*p* = 0.00264	1.3 ± 1.2	1.2 ± 1.2	NS
Nocturia ^b^	2.7 ± 0.9	2 ± 1.3	*p* = 0.00256	2.4 ± 0.9	2.5 ± 0.9	NS
Times/nocturia (episodes) ^b^	2.3 ± 1.4	1.1 ± 1.1	*p* = 0.0001	1.9 ± 1.1	1.9 ± 1	NS

^a^ T1 next visit after changes in treatment. ^b^ Wilcoxon signed rank test. Level of significance in *p* < 0.01. Abbreviation: NS: no significant; SCOPA-AUT-U: scale for outcomes in Parkinson’s disease for autonomic symptoms in urinary symptoms.

## Data Availability

The data presented in this study are available on request from the corresponding author. The data are not publicly available due to private patient´s data.

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
