# Peer review of "SURINPARK: Safinamide for Urinary Symptoms in Parkinson’s Disease"

_brainsci, 2021, doi:10.3390/brainsci11010057_

Round 1
Reviewer 1 Report
SURINPARK: safinamide for urinary symptoms in Parkinson´s disease
brainsci-1054244
An interesting report.
32 with safinamide and 61 control.
SCOPA-AUT-Urinary subscale showed improvement in safinamide group.
My comment:
You added MAO-B inhibitor safinamide to MAO-B inhibitors rasagiline 9% and selegiline 1%: why you added? Should be explained in method/discussion section from view point of ethics and science.
You should cite: Rasagiline effect on bladder disturbances in early mild Parkinson's disease patients, 2014 Brusa L et al.
Minor: tenesmus is not relevant to bladder, delete.
Author Response
Dear reviewer,
We appreciate your most valuable suggestions and corrections for our study.
Point 1. You added MAO-B inhibitor safinamide to MAO-B inhibitors rasagiline 9% and selegiline 1%: why you added? Should be explained in method/discussion section from view point of ethics and science.
We have included a sentence clarifying that when safinamide was prescribed to a patient taking MAO-b inhibitors (rasagiline or selegiline), these were withdrawn when safinamide was started.
Point 2. You should cite: Rasagiline effect on bladder disturbances in early mild Parkinson's disease patients, 2014 Brusa L et al.
We have also included in the references the study you suggested in your review (Brusa, 2014)
Point 3. Minor: tenesmus is not relevant to bladder, delete.
Unfortunately, as tenesmus is part of the SCOPA-AUT scale, we can not exclude this symptom from our analysis.
Best regards,
Ana Gómez López

Reviewer 2 Report
The authors have correctly identified an area of need as management options are limited for urinary dysfunction in PD. The research question has been set as "to assess the possible clinical effect of safinamide on improving the urinary symptoms of PD patients in daily clinical practice." They have used retrospective data collection and SCOPA aut for answering this question.
The methods applied to answer the question should have included a prospective open label or possibly a randomised control study with appopriate use of urodynamics or other bladder investigations. This is my main criticism of this research plan but I understand the authors cannot correct this now.The authors acknowledge the study limitations but have not provided any justification of why they chose not to use standradised urological questionnaires or urological investigations. If this was a part of another safinamide study which could justify the study design as a post hoc analysis from a data set generated for another clinical trial (for example).
The authors acknowledge the study limitations in the discussion but indicate that "inclusion of consecutive patients helps reduce the potential biases ". This is not true in my undertanding as biases still remain.
I am happy with the conclusion section of the article but it is common to be misled when reading the conclusion on abstract. Maybe authors can give a clearer consideration to the limitations mentioned in conclusion section of the article and consider reconsidering the current conclusion line of abstract "Safinamide could be helpful in the improvement of urinary symptoms in PD"
Author Response
Dear reviewer,
We appreciate your most valuable suggestions and corrections for our study.
Point 1. The methods applied to answer the question should have included a prospective open label or possibly a randomised control study with appropriate use of urodynamics or other bladder investigations. This is my main criticism of this research plan but I understand the authors cannot correct this now. The authors acknowledge the study limitations but have not provided any justification of why they chose not to use standardised urological questionnaires or urological investigations. If this was a part of another safinamide study which could justify the study design as a post hoc analysis from a data set generated for another clinical trial (for example).
We agree that a prospective design and urodynamic study or specific urological tests would be best to clarify the questions we raised. However, this was a retrospective pilot study, and unfortunately, we lack the funding and urologic support to further develop this kind of design. We would like also to clarify that we are not developing a more complex study on this issue at the moment.
Point 2. The authors acknowledge the study limitations in the discussion but indicate that "inclusion of consecutive patients helps reduce the potential biases ". This is not true in my undertanding as biases still remain.
We have removed the sentence in our discussion "inclusion of consecutive patients helps reduce the potential biases" according to your suggestion.
Point 3. Maybe authors can give a clearer consideration to the limitations mentioned in conclusion section of the article and consider reconsidering the current conclusion line of abstract "Safinamide could be helpful in the improvement of urinary symptoms in PD"
We have rephrased the conclusion according to your suggestions
Best regards,
Ana Gómez López
